# "Live a normal life": Constructions of resilience among people in mixed HIV status relationships in Canada

Minhui Yang[1], Amrita Daftary[2], Joshua B. Mendelsohn[3], Molly Ryan[1], Sandra Bullock[1]\*, Laura Bisaillon[4], Adam Bourne[5], Bertrand Lebouché[6], Tamara Thompson[7], Liviana Calzavara[1]

1 Dalla Lana School of Public Health, University of Toronto, Toronto, Canada, 2 School of Global Health, and Dahdaleh Institute of Global Health Research, York University, Toronto, Canada, 3 College of Health Professions, Pace University, New York, New York, United States of America, 4 Department of Health and Society, University of Toronto Scarborough, Toronto, Canada, 5 Australian Research Centre in Sex, Health and Society, La Trobe University, Melbourne, Australia, 6 Faculty of Medicine, Division of Infectious Diseases and Chronic Viral Illness, and the Research Institute of the McGill University Health Centre, McGill University, Montreal, Canada, 7 Faculty of Health Sciences, Douglas College, Coquitlam, Canada

* s.bullock@utoronto.ca

**Data Availability Statement:** De-identified data excerpts are presented within the paper, as a minimal dataset. There are ethical restrictions on sharing the complete de-identified study

## Abstract

Positive Plus One is a mixed-methods study of long-term mixed HIV-serostatus relationships in Canada (2016–19). Qualitative interviews with 51 participants (10 women, 41 men, including 27 HIV-positive and 24 HIV-negative partners) were analyzed using inductive thematic analysis to examine notions of relationship resilience in the context of emerging HIV social campaigns. Relationship resilience meant finding ways to build and enact life as a normal couple, that is, a couple not noticeably affected by HIV, linked to the partner with HIV maintaining viral suppression and achieving "undetectable = untransmittable" (U = U). Regardless of serostatus, participants with material resources, social networks, and specialized care were better able to construct resilience for HIV-related challenges within their relationships. Compared to heterosexual couples and those facing socioeconomic adversity, gay and bisexual couples were easier able to disclose, and access capital, networks and resources supporting resilience. We conclude that important pathways of constructing, shaping, and maintaining resilience were influenced by the timing of HIV diagnosis in the relationship, access to HIV-related information and services, disclosure, stigma and social acceptance.

## Introduction

The past decade has seen a critical shift in clinical and public health responses to human immunodeficiency virus (HIV), with antiretroviral therapy (ART) enabling people in many settings to access high levels of viral suppression [1]. While earlier in the HIV/AIDS pandemic we might have distinguished between people being "HIV-positive" and "HIV-negative", today we refer to their treatment status, "on treatment" or "not on treatment" [2]. Priorities have

transcripts, due to the ability of one person to identify their partner's statements. The study consent letter included the statement: "Although we may use quotes from your interview in future publications or presentations, all quotes will be anonymous. Also, each quote will be general enough in nature that you will not be identifiable, even by your partner." Providing fully-transcribed interviews would allow one partner to easily identify their partner, and possibly other people. As a result, the data can only be made available to other researchers if terms of anonymity are met via data transfer agreements with the corresponding author.

**Funding:** This research (all authors) was supported by the Canadian Institute of Health Research (CIHR, grant number: MOP-137009), https://cihr-irsc.gc.ca/e/193.html and the Social Research Centre in HIV Prevention (SRC). The funders had no role in study design, data collection and analysis, decision to publish, or preparation of the manuscript.

**Competing interests:** Amrita Daftary is a section editor for PLOS Global Public Health and academic editor for PLOS One. Bertrand Lebouché has received grants for investigator-initiated studies from ViiV Healthcare, Merck, and Gilead; consulting fees from ViiV Healthcare, Merck, and Gilead. All other authors have no competing interests.

turned to connecting people with HIV with early ART to facilitate viral suppression. Correspondingly, "treatment as prevention" (TasP) and "undetectable = untransmittable" (U = U) have emerged as key prevention approaches, and to manage the risk of transmission [2–4].

Post-HIV-diagnosis resilience has become a topic of increasing relevance with this shifted framing of HIV, as a once fatal disease to now a long-term, manageable health condition [5]. Resilience is typically defined as a "dynamic process encompassing positive adaptation within the context of significant adversity" [6] characterized by "normal development under difficult conditions" [7]. For people living with chronic illness, resilience is further specified as the capacity to negotiate life despite the adversities brought about by long-term suffering [8]. Resilience within couples can have fluid and multi-dimensional constructions, depending on the cultural and material contexts in which it is developed; such as: level of trust, the differences in economic position and power between couples, cultural understandings about HIV and about relationships [5, 9, 10]. For mixed HIV-status couples, resilience is found to be closely bound to HIV treatment and prevention [11]. Research shows that partners can develop a sense of connected or shared responsibility of living with HIV because of their emotional connections and commitments to each other [2]. Partners may adhere to ART not only for the health of the HIV-positive partner, but also out of mutual interest in preventing transmission to the HIV-negative partner and protecting the quality of their shared life [12, 13]. Attaining viral suppression via adherence to HIV treatment and prevention in this way could allow keeping life as normal for both HIV-positive and HIV-negative partners, and substantially comprise their resilience. Even so, asymmetric pressures and responsibilities of dealing with HIV-related challenges can arise among partners, shaping diverse presentations of resilience within mixed status relationships [10, 14, 15]. Recent work from Australia suggests that mixed HIV-status couples may also face distinct experiences within their home as compared to the public domain [16].

Building on this work, the Positive Plus One study sought to explore resilience among mixed HIV-status couples in Canada where HIV is increasingly experienced as a long-term health condition in the era of U = U [17]. To achieve this goal, we used in-depth interviews to assess factors influencing resilience among 51 people from 32 mixed-status couples enrolled in the Positive Plus One study. We see their lived experiences and views on benefits and challenges related to their mixed HIV status as central in efforts to promote the health and wellbeing of couples living under similar circumstances [2].

## Methods

Positive Plus One was a national mixed-methods study of mixed HIV-status couples undertaken between 2016 and 2019 in Canada. Detailed study methods have been published [11, 18]. The main study sample comprised 613 adults from 10 provinces engaged in a current or recent (past two years) relationship of three months or longer, where one partner was HIV-positive and the other HIV-negative, having disclosed their HIV status to one another. Upon completion of the survey, participants were invited to volunteer for a more in-depth telephone interview, of approximately 1 hour duration. This is one of two qualitative papers from the study, and focuses solely on the results pertaining to resilience [11].

### Recruitment

Participants were recruited via 143 AIDS Service Organizations and other non-governmental organizations and 35 Infectious disease and general health clinics, via direct contact with staff and physicians, and indirectly via posters and pamphlets, word-of-mouth, media articles, and social media. Six hundred and thirteen participants completed a structured survey

(telephone or on-line), the findings of which are published elsewhere [11, 14, 18]. Survey participation was voluntary, people entered the online web survey, or called the office to take-part by telephone, only if they were interested in taking part in the study; therefore, we have no knowledge of the number people in mixed HIV-status relationships choosing not to participate, nor their reasons for non-participation. After completion of the survey, 51.7% of the participants indicated interest in volunteering for an in-depth audio-recorded telephone interview. Interested participants were asked to provide their contact information–email was preferred by 54%; phone by 29%; and 17% indicated both or either. For the 17% who indicated both/either contact method, three attempts were made by telephone prior to contacting their email addresses.

Purposive recruitment from those volunteering for the in-depth interview was aimed at maximizing participant diversity on the dimensions of race, age, gender, sexual orientation, length of living with HIV, and relationship type. Of the volunteers, 69 were purposively selected for interview; 51 were ultimately interviewed prior to reaching data saturation. Two people (3%) who were approached were no longer interested–they reported being too busy; and 16 people (23%) were unable to be contacted/located at the time we were reaching out by email and telephone to schedule interviews, prior to reaching data saturation. These 18 non-participants represented a broad range of survey participants. Ten reported being HIV-positive (nine of which were on ART) and eight were HIV-negative. Fifteen people were in current relationships (ten with a partner in the study, five without a partner in the study) and three were in recent mixed-status relationships. Three people reported low relationship satisfaction. Twelve identified as male (seven identified as gay, three heterosexual, one bisexual, and one not-identifying). Six identified as female (five identified as heterosexual, one bisexual). Thirteen people identified their race/ethnicity as white, with five identifying as visible or racialized minorities.

## Study participants

The sub-group of participants purposively recruited from the volunteers into the qualitative study ensured inclusion of HIV-positive and -negative people reporting different levels of relationship satisfaction, from current and recently ended relationships, in geographic locations across Canada (rural and urban residence), and having experienced varied relationship formats (single and multiple partners). These participants have been described in detail elsewhere and are summarized here [11]. The 51 interview participants included both partners in a mixed-status relationship, where possible (n = 38; 19 HIV-negative, 19 HIV-positive), one partner in a current relationship (n = 9; 3 HIV-negative, 6 HIV-positive), and people who had previously been in a mixed-status relationship (n = 4, 2 HIV-negative, 2 HIV-positive).

The median age of the 51 participants was 43 years (range 23 to 69 years), of whom most identified as men (n = 41). People with HIV comprised 53% of the sample and they had been diagnosed between six months and 32 years (median = 10 years) prior to the interview. Two-thirds (n = 35) of the sample self-identified as gay or bisexual, 15 as heterosexual, and one person did not specify a sexual orientation. Thirty-one (61%) participants indicated being in a monogamous relationship. About a quarter (n = 13) said they were visible or racialized minorities (seven were recent immigrants to Canada). Over 70% (n = 37) were employed. Twenty-nine (91%) relationships reported a virally suppressed partner, and one reported a partner not on ART. The duration of relationships ranged from three months to 26 years (median = 5 years). In 20 (63%) relationships, HIV was diagnosed prior to the start of the relationship. Seven (22%) couples received the diagnosis during the relationship and remaining 15% (n = 5) received their HIV diagnosis just as the relationship started.

## Interview

Data were obtained by telephone interview. When speaking with the participant to set the appointment for the interview, participants were advised to ensure that they would be in a private location where no one was able to overhear, and when their partner would not be present. Prior to obtaining verbal informed consent, interviewers asked participants if they were alone in a private location where they would not be overheard or interrupted. Informed consent was then obtained prior to interview start. The study qualitative interview guides are provided in an online Appendix (S1 Appendix). The four study interviewers were selected to represent key demographic groups of participants. One interviewer was known to be HIV-positive and had previously been in an HIV mixed-status relationship; three were female, one male; two identified as heterosexual, one bisexual, and one was gay; and each of them had worked in the area of HIV research for several years.

To assess data saturation, the research coordinator listened to each interview as it was returned to the office following completion, to assess for repeated responses to questions. Interviewers also met with the research coordinator biweekly to review enrollment, discuss themes emerging, and discuss data saturation. Data collection was completed once all parties agreed that responses were routinely repetitive, and no new information was presented.

## Analysis plan

Interview recordings were transcribed verbatim, French interviews were translated and back-translated twice for accuracy. Personal identifying information was redacted, and participants were assigned a pseudonym to safeguard anonymity. Due to the exploratory nature of the study, we chose an inductive thematic analysis methodology for the data analysis. We concurrently picked up on coping strategies that mixed-status couples engage in to take care of themselves. This led us to adopt resilience as a theoretical basis from which to enhance our interpretations of data pertaining to couple's means of dealing with HIV and maintaining mutual wellbeing in the contexts of their relationships. Accordingly, we refined our interpretations around inductively emergent codes and concepts with published frameworks of resilience applied to understand experiences of people living with HIV as well as couples affected by HIV [19, 20].

Transcripts were coded by two research team members (MY, MR) who led the thematic analysis using an inductive approach, identifying patterns within and across data to generate emerging themes [21]. MY and MR performed parallel data coding using Dedoose desktop software followed by meetings to consolidate coding. Initial themes evolved through research team discussions (AD, JM, SB, LC). Researcher memos were valuable for helping to see patterns in the data as expressions of reflexive practice [22].

In the first round of reading and coding of the 51 interviews, we identified concepts related to sexual and intimate relationships (18 codes), HIV disclosure (9 codes), family relations (5 codes), experiences in health care (15 codes), diagnosis (6 codes), medication (9 codes), transmission (7 codes), social support (21 codes), knowledge and attitudes about HIV (17 codes), and lived experience (11 codes) as meaningful aspects of mixed-status relationships for study participants. These allowed us to follow through on several lines of inquiry, as is common in qualitative analysis. We identified key concepts that spoke to the biomedicalization of HIV in the everyday lives of couples, which we have scrutinized distinctly and published elsewhere [11]. We concurrently picked up on strategies and resilience that mixed-status couples engaged in to take care of themselves. It is these concepts that we explore in further detail within the current paper, as they are crucial to understandings about how partners cope as a couple, following an HIV diagnosis or disclosure. While participants in our study did not use

the word "resilience," we understand how and what people do for their mutual wellbeing as forms of resilience, and thus the conceptual grounding for this analysis.

## Research ethics

All methods for this study were carried out in accordance with relevant guidelines and regulations. This study received ethics approval from the University of Toronto research ethics board (REB) (Protocol 31855) [18]. Given involvement in active recruitment, yet no involvement in participant consent or data collection procedures, only a few of the health/HIV clinics judged it necessary to obtain approval from their own REBs. The study underwent review and obtained approval from REBs at McGill University (2017–1779, 16-035-MUHC, eReviews_5368), University of Saskatchewan (15–399), St. Michael's Hospital (16–343), Toronto Public Health (2016–02), Nova Scotia Health Authority (NSHA REB ROMEO FILE #: 1022121), Prince Albert Parkland Health Region (no REB number), and Regina Qu'Appelle Health Region (REB-15-133).

All research team members and staff signed an oath of confidentiality. All participants indicated their informed consent following completion of eligibility screening to participate in the initial survey–either verbally if completed by telephone or by clocking the "I consent" box if completed online. The survey program did not retain screening data if consent was not provided. Further telephone verbal consent was required for participants to complete the in-depth qualitative interview, providing permission to publish anonymous quotes from their interviews. Consent of both partners within an HIV mixed-status relationship was required to link their responses; and participants' responses were not shared with their partners even when their data were linked for analysis. Survey and qualitative data were anonymous, and unlinked contact information was provided to receive a token of appreciation and to volunteer for the qualitative interview; and was deleted once it was no longer required for these purposes. All anonymized data will be kept for seven years post project completion.

## Results

This research occurred during a period of rapid acceptance of messaging centered on U = U in Canada [1, 17, 23], a concept that has reshaped perceptions of HIV and associated stigma but, as our study shows, is not yet universal or sufficient to quell concerns among all groups of people. In-depth interviews with participants engaged in mixed-status relationships across the country point to a shared understanding of resilience as the capacity of keeping life as normal. We uncovered how pharmaceutical technologies (i.e., current ART for achieving "treatment as prevention", and the use of pre-exposure prophylaxis), access to information and services, ease of disclosure, and social acceptance, both enabled by and enabling achievement of U = U. These factors contribute to shaping the constructions of normalcy and, in turn, not only affirm couples' resilience but also serve as a gateway for expanded relationship resilience.

Resilience for mixed-status couples in our study centred around their strategies and resources to "live a normal life" and "like a normal couple", that is, as life not noticeably disrupted by HIV. We unpacked how this normal life narrative intersected with resilience, as we have critically engaged with the general processes of normalization elsewhere [11]. We found that resilience for mixed-status couples began with their capacity to support the HIV-positive partner in reaching viral suppression. Strategies to achieve suppression differed for persons with HIV and their HIV-negative partners and varied by their understanding and acceptance of U = U as it related to their lives, sexuality and sexual practices, and mental and physical health. Participants' stories also illustrated how their resilience was impacted by intra-relationship, interpersonal, and social factors. Noticeably, compared with heterosexual couples, we

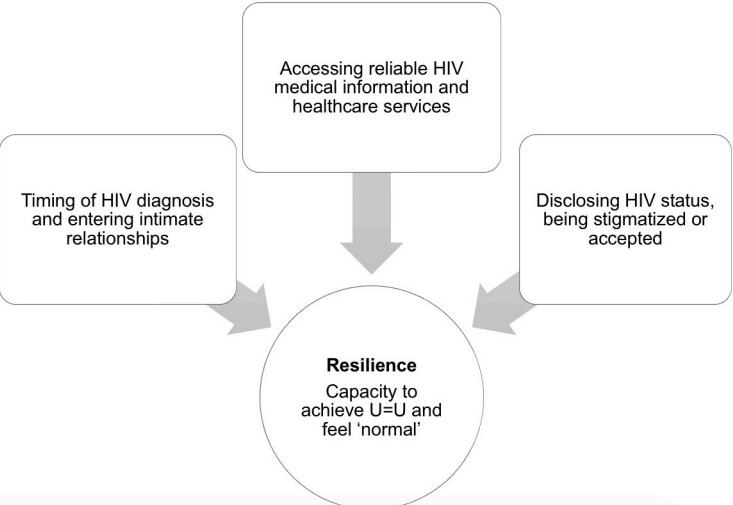

**Fig 1. Factors affecting the resilience of HIV mixed-status relationships.**

found that gay couples were more likely to obtain support from local LGBTQ2S+ communities and corresponding resources. Fig 1 highlights how three factors put forth by mixed-status couples in our study–the timing of HIV diagnosis within the relationship, access to HIV information and healthcare, and outside stigma and acceptance upon disclosure–challenged the construction and maintenance of resilience in their relationships. These factors were prominent in our four thematic findings and are elaborated upon ahead.

## Resilience: Capacity to achieve U = U and feel "normal"

Participants showed a common understanding of HIV treatment and prevention, supporting a mutual desire to engender normalcy in their relationship. When describing this normalcy, it was a state where HIV did not intervene in their routine, everyday lives and often meant that the HIV-positive partner was clinically well as per their viral load, there was little possibility of transmitting HIV, and they were free not to define themselves through their HIV status. Achievement of these goals by the HIV-positive partner facilitated enjoyment of a relationship unencumbered by HIV.

> It's very normal, it's just one pill I have to take when I woke up in the morning. . .I'm undetectable—actually, I have really good health.
>
> *(HIV-positive gay man)*

This conceptualization of resilience was connected with prevailing understandings of U = U (i.e., when HIV-positive people's viral load is undetectable, they are not able to sexually transmit the virus to others) and TasP (i.e., taking ART can prevent sexual transmission by fully suppressing the virus) [23]. Participants became familiar with these messages through media, healthcare providers, and AIDS service organizations. They interpreted these messages to mean an undetectable viral load was the way to clinical health for the HIV-positive partner, sexual safety for the HIV-negative partner, and more sexual pleasure with fewer worries about transmission between partners. U = U was seen to facilitate many of their perceived prerequisites for normalcy within a relationship.

It means that we no longer feel like people, including myself, have felt for a very long time, like walking sexually transmitted diseases. We felt infectious. We felt, you know, that HIV it defines us, you know, it's sort of who we are. But now with U = U it's like, no, that's not who I am. Like anyone else who has, you know, a chronic infection that they can't pass on, you know, it doesn't define their lives, right? So, it's transformative.

*(HIV-positive heterosexual woman)*

HIV-positive and -negative participants alike reported how acceptance of U = U reduced anxiety about sexual intercourse with their partner. Many felt confident not wearing condoms or taking pre-exposure prophylaxis if viral suppression had been achieved. Condoms, in particular, had affected intimacy for several participants who cited their use, or discussions about their use, as a trigger for relationship turmoil and break-up. For most people we talked with, protection offered by U = U and TasP strategies eliminated the need to continue these preventive practices and served to enhance sexual compatibility.

I wish we'd been told about U = U. I've been undetectable for many, many years at that point and so we didn't need to use condoms, but we didn't know. We weren't told. So if we'd been told the correct information, I'm not sure it would've saved our relationship but it certainly would've made it easier.

*(HIV-positive heterosexual woman)*

Now we don't really manage the risk anymore because we do everything we want without protection. We feel there isn't really any risks for us.

*(HIV-positive gay man)*

U = U bolstered resilience by infusing safety and equilibrium, facilitating mixed-status couples to have a renewed focus on other, perceptibly normal, aspects of relationships. The achievement of U = U also became a sign of evaluating their own resilience. Although U = U and TasP were central to how partners described their sense of normalcy, several other factors were also seen as relevant their collective resilience: when HIV entered the relationship; access to medical information and health care services; and disclosure of the relationship, stigma and/or social acceptance.

## Timing of HIV diagnosis and entering intimate relationships

The timing of the HIV diagnosis shaped resilience building among couples and the overall trajectory of the relationship. When both partners entered a relationship with full knowledge of the presence of HIV in one partner, they developed strategies to manage HIV-related challenges more smoothly than when diagnosis occurred during an ongoing relationship. This is largely because partners who had lived with HIV had already developed ways of managing HIV before entering the relationship. Many who benefited from pre-existing living and working in environments that accepted their HIV-status, and who had cultivated strong social relations with friends and family, were better able to cope with HIV independent of a partner. Experiences and environments that supported normalizing their HIV-positive identity outside of the relationship bolstered their individual resilience and allowed HIV to bear less influence on everyday couple dynamics within new relationships.

HIV-negative participants also sensed when their partners had already developed their coping strategies. When the positive partner played the role of educator and advocate, and shared their knowledge of HIV, including access to key information, resources, and networks,

the negative partner often expressed relief, feeling more secure about their partner's wellbeing, and less need to directly engage with those experiences or become a guardian of their health or emotional state. These practices helped to solidify a sense of normalcy, which contributed to the development of the couple's resilience.

> I also get him, my partner, lots of information that's on social media, that I read on social media and from the website. . . sometime I saw the news. . . there's some information online. . . When I saw those, I will forward to my partner and then we will discuss.

> *(HIV-positive gay man)*

Couples that dealt with an HIV diagnosis during their ongoing relationship had comparatively more difficult experiences. In these cases, HIV-positive partners had not already developed individual resilience; they were faced with managing news of their diagnosis and risk of HIV transmission. Both HIV-positive and -negative partners experienced a stage of under-preparedness and a break from normalcy; each played an active role in reparation and resilience building. Several HIV-negative partners recalled enduring a period of intense struggle, ambivalence, and emotional distress as they came to terms with the diagnosis, and reconciled concerns about the clinical health of their positive partner and their own health risks. Partners turned to each other for support and guidance around next steps. In some instances, however, pressures triggered the couple to turn away from each other. For couples that survived, normalcy returned gradually, after the hurdles of HIV disclosure and acute HIV management were overcome, and when the HIV-positive partner achieved U = U. Sometimes, longer periods were needed for couples to accept and adapt to living with HIV.

> One of the challenges that I mentioned to you as well was with me being scared, like, when we kiss or when we have sex. Those things are getting me scared that—and how we overcame that was she went and got treatment, she's undetectable, the doctor gave me—gave me information on how people are undetectable. . . like one person negative, one person positive, how they would have sex openly but it would not be—it's not getting transmitted and the other person keeps on using their medication. So that gave me more confidence.

> (*HIV-negative heterosexual man*)

Resilience was thus constructed in unique ways depending on the sequence of diagnosis and relationship onset. Partners who discovered HIV, essentially together, during the course of their relationship tended to experience simultaneous uncertainties followed by a period of mutual reliance and co-dependence in the management of HIV-related queries and challenges. On the other hand, HIV-positive partners who entered relationships after knowing about and learning to manage their illness tended to be more independent, and shoulder responsibility in the management of HIV-related concerns in the relationship.

## Accessing reliable HIV information and services

Access to reliable medical information and health care were understood to be essential for achieving and accepting U = U, preserving sexual safety, and enabling the re-construction of normalcy (among partners who learned about HIV together) or preservation of normalcy (among partners who entered relationships with knowledge of their mixed HIV-status). From the perspectives of HIV-positive partners, this meant sustained and trusted access to HIV clinical care including medical checkups, ART refills, viral load testing, and even participation in

research. From the perspective of HIV-negative partners, this meant an assurance that their partner was receiving the required clinical attention to maintain an undetectable viral load.

> My partner takes his medication every morning and every evening, I know he doesn't fall into–he doesn't become–and he is always, like I said, he gets checked up every 3 months for it so I am not afraid to get a bad surprise, where he wouldn't be undetectable.

> *(HIV-negative gay man)*

Participants' access to the most up-to-date information and medical care for HIV was contingent on social and economic contexts. Financial barriers, for example, were only partly alleviated by the public health system that offered basic HIV treatment services at no charge but limited access to allied healthcare services such as prescription medication, physiotherapy, mental health and addiction services, etc. Specialized supports and newer treatment options required private insurance (typically job-related) or out-of-pocket payment. While a single-payer healthcare system funds some HIV care services for all Canadians, including ART, provinces and territories maintain distinct lists of eligible ART medications, and observe distinct eligibility criteria for national and provincial subsidies for allied health services including for prescription fees. Several participants living outside of urban areas with fewer specialized facilities reported barriers to HIV care and ART.

> [Treatment] needs to be brought in quicker, I think. We obviously–we both work part time so we don't have health care coverage through our jobs. He's being funded through the [foundation name] for his medication because there's no way we could afford his medication on a monthly basis. And we had to wait–like, from the time of diagnosis until he started treatment was almost six months. . . the doctor and the specialist calling and saying, 'We need this funding in place before we can start treatment. We need it in place. And if he doesn't get it, he's going to die.'

> *(HIV-negative heterosexual woman)*

> We only have one infectious disease doctor in [province name]. . .I know there's like 300 HIV patients alone, and we only have one doctor. . .We had to fight to get that doctor. We never had an infectious disease doctor here for two years.

> *(HIV-positive heterosexual woman)*

Participants discussed worries about HIV service interruption. Those who had third-party private insurance plans covering all ART costs worried about what a change in their job, and insurance status would mean for their long-term health. Some participants indicated that disruptions to HIV clinical care on account of costs had inserted tensions into their relationships–a concern experienced by HIV-positive partners regardless of who the primary holder of their insurance policy was.

> Every month there's a bit of a strain on our financial relationship because my drugs are expensive, and I'm self-employed, so I don't have a benefits package anymore. [Partner's name] does. He has a public service benefits package, so that covers a great deal of the cost. But every month we're aware that my meds are non-negotiable. There have been times, probably early on in our relationship, where we had to recognize that my meds may have to take precedence over a loaf of bread.

> *(HIV-positive gay man)*

Social networks that facilitated HIV-positive partners' access to advanced HIV treatments including positive relationships with healthcare providers were also important drivers of clinical and psychological wellbeing and supported the desired level of stability and resiliency to HIV within relationships. For example, participants who worked at AIDS service organizations, healthcare facilities, or had personal links with providers, community advocates, and other people with HIV were privy to the most current information about HIV treatment and prevention, and insider opportunities to advocate for access to them. Indeed, several participants' primary source of access to novel or cutting-edge ART regimens were through enrollment in randomized controlled trials of experimental therapies, learned of through personal connections. Good relationships with medical providers meant regular access to up-to-date medical information, tailored supports to meet their individual needs and priorities, and ample opportunities to receive supportive services.

> I am lucky in the sense that I have my cousin who introduced us is a nurse and he actually used to work in the HIV and STI clinic in [city name], so he's a good wealth of information, and one of my other best friends is also a nurse. So, I have them as kind of information as well.
>
> *(HIV-negative gay man)*
>
> If something's wrong medically, I'll talk to my friends who are nurses, and then I also have a very good relationship with my ID [infectious disease] doctor. I actually work at the hospital where I go see my ID doctor so like, if he sees me in the hallway, because I only go every six months to the doctor. . .he actually does like a mini check-in with me. "Hey, how's it going? How are things?" So it's nice to have that little support there as well.
>
> *(HIV-positive bisexual woman)*

By contrast, other participants shared their disappointment with healthcare providers who treated them in stigmatizing ways. One participant ended up accessing multiple facilities for information and guidance, to avoid having to communicate with the primary physician who had prescribed her ART:

> I use my family doctor, who I was reluctant to use initially. He had made a not-so-kind comment to my husband when my husband said that I was positive. He just kind of made this off-the-cuff remark of, 'Oh my gosh, I hope you use protection.' . . .There were things initially when he was my family doctor that I didn't feel comfortable talking to him about, and so I would go to a walk-in clinic and then lie, because I wouldn't tell them that I was positive.
>
> *(HIV-positive heterosexual woman)*

Access to up-to-date information and quality HIV clinical care services was not only associated with structural factors such as health insurance coverage and distribution of specialized medical resources, but also influenced by personal networks of the mixed-status couples. Participants sought to be treated by providers who involved them in healthcare decisions and adopted holistic and inclusive approaches to service provision. Access to new knowledge and HIV care was sometimes the result of who the person knew, where they were located in the system, and via personal, local, and provincial resources. Feelings of being marginalized were exacerbated for many participants who were either unable to access supportive providers or had few personal and professional networks to support them in their challenges related to sexual health, HIV clinical care and, relatedly, intimate relationships.

## Disclosing HIV status, being stigmatized or accepted

Most HIV-positive partners grappled with HIV disclosure starting from the moment of diagnosis. Decisions to disclose HIV were mired in fears of stigma and experiences with discrimination. Participants tended to disclose to individuals with whom they had an established level of trust and/or individuals who they felt would not judge or reject them, such as very close confidantes in their family or social networks, others living with HIV, or others perceived to be understanding of or accepting of people living with HIV. Some participants felt that HIV stigma had declined based on their exposure to information around ART, U = U, and transformation of HIV into a manageable condition. Even so, all participants said that disclosure to loved ones had almost never been easy.

Participants' disclosure encounters, stigma, and acceptance within their relationships were guided by the timing of HIV diagnosis in the relationship. In this study, all HIV-positive participants who entered their relationships with knowledge of their HIV status reported disclosing to their intimate partner before the first sexual act. They felt obliged to disclose despite the risk of rejection, describing it as the "right" thing to do. Participants who learned about HIV during an ongoing relationship similarly felt compelled to disclose to their partner almost immediately after learning their HIV test result. In both cases, the disclosure event was worrisome; the disclosing partner always worried about rejection. Those who were newly diagnosed simultaneously processed the reality of having HIV, the meaning of their diagnosis for their wellbeing, and how they might have acquired the infection. HIV-negative partners worried about HIV risk and protection, betrayal (even while recognizing events that lead to HIV acquisition likely preceded their relationship), and/or worry and compassion for their partner.

> I didn't know that he is HIV positive at the beginning but the first time we met each other, he told me before to have sex. He told me that he is HIV positive. My reaction, I'd seen it as normal. I just say it's okay, just we have to use protection. But it was my reaction but I was, I think, very scared.
>
> *(HIV-negative gay man)*

When disclosure was proximal to diagnosis, simultaneously learned by both partners in an established relationship, and not linked to an infidelity, it served to strengthen partners' connection to each other and commitment to the relationship after a temporary period of uncertainty and stress. Relationships that did not survive disclosure or the temporary upheaval, however, quickly fell apart.

Disclosure also encompassed discussing the relationship with couples' personal networks and healthcare providers. All participants assumed this was the HIV-positive partner's decision to make. HIV-positive partners felt disclosure outside of their relationship, to people who were not close confidantes, was unnecessary and could invite undue attention and even discrimination. Disclosure could also out them as belonging to a relationship that was different, challenging the valued perception of normalcy. This was seen as particularly disruptive for mixed-status couples who were undergoing other major life changes. For example, one couple discovered their HIV status in the process of immigrating to Canada, at once facing stigma, anxieties of adapting to new environment, and a lack of social support:

> I'm scared of what people are going to say. And if you're not HIV or you're not in that—working in that field, you... just listen to what you hear... Basically, I don't think I can tell people, because all of based on what I'm hearing, it's not only [country name]. All over people stigmatize these days, but they sell it more different here in Canada.

*(HIV-positive heterosexual woman)*

Among other participants, we saw HIV had become normalized, similar to other long-term health conditions, and especially among participants who identified as part of the gay, bisexual, and other men who have sex with men (GBMSM) community. Same-sex sexual rights movements were part of everyday conversations. Discussions about HIV, ART, prevention and related public health interventions were commonplace; the idea of taking pills for treatment was increasingly likened to taking pills for prevention. Both participants living with HIV and HIV-negative participants of GBMSM communities were able to talk about HIV, their HIV status, or mixed HIV couple status with a level of openness that was not described among participants engaged in heterosexual relationships.

> I actually live very close to [city's] gay village. I think probably of all the places in Canada, they're probably the most well-versed. My friends have all educated on these sorts of things. I think the interesting thing about that is that we all have, we're all in the same pool, essentially, of information and situations and so the information is really pooled around.

*(HIV-negative gay man)*

> He's also on PrEP as well after he met me. . .when it's become very popular in [province] now, starting September/August of last year. And so, when I talk about taking my pill, he's also taking a pill as well for himself.

*(HIV-positive gay man)*

Beside the comfort of belonging to the GBMSM community, some participants reflected that the outpouring of support and absence of negative reactions was further connected to privileges brought on by their socioeconomic status and related social capital.

> I think the biggest problem here in [province name] isn't for me and my partner. It's for the poor, the homeless, the drug users. And I do notice that. And I know, myself, being, you know, a lawyer, and having money and wearing a suit to the doctor's office, I am being treated differently than some of the others. If that makes—you know? And my partner notices that as well. . .

*(HIV-positive gay man)*

This contrasted with participants who were living in settings with less exposure to, and less acceptance of, people living with HIV and for whom discussions about HIV and disclosure were heavily guarded. Disclosure outside of the relationship was reserved for allies of the HIV-positive partner who had been involved in their lives well before the relationship. New disclosures were crucially avoided.

> So, I tell my sister and my husband. . . I don't think I'm at that stage to disclose to anybody, family. In [country], people stigmatize a lot. . .no one is confidential. . .Even if I was sick and I was diagnosed at the hospital, everybody would have known. . .I'm here and I still can't, I have so many friends but I don't think I can disclose.

*(HIV-positive heterosexual woman)*

Participants who identified as gay were aware of these prejudices that prevailed outside of their existing circles:

Many friends of us knew about it throughout the years and they were very, very supportive. But not all my friends. I know that other people that I don't feel comfortable talking about. It is not something that you go and tell everybody.

*(HIV-positive gay man)*

Disclosing and accepting the HIV status of positive partners was therefore key to the relationship resilience for most couples who participated in our study. Moreover, disclosure of mixed HIV status to social networks helped many couples acquire additional needed supports. However, despite the ongoing public normalization of HIV enabled by the new paradigms of U = U and treatment as prevention, stigma and discriminations to PLHIV perpetuated in some micro interpersonal environments. Couples who felt this stigma in their close circles tended to avoid disclosure, which introduced more barriers to accessing the resources needed for constructing resilience.

## Discussion

Several factors affected resilience among the mixed-status couples represented in our study: timing of the entry of HIV into relationships, access to health information and services, and experiences of HIV disclosure, stigma and acceptance. These themes reinforced a core discourse that focused on sustaining everyday normalcy through minimized intrusion of HIV-related challenges within the routine functioning of relationships. Within the U = U paradigm, undetectable viral load was key to the HIV-positive partners' health, while also ensuring the HIV-negative partner remains free from HIV infection. To be undetectable by adhering to ART and maintaining viral suppression signified to couples that they could enjoy a normal life just as other couples did. Undetectable viral load, therefore, was significant beyond HIV prevention by minimizing disruption to relationship functioning and the need to spend emotional resources on managing HIV-related challenges. As a result, relationships could be re-centered on trust, responsibility, and love.

In Australia, Philpot and colleagues [16] identified distinctions between the normalcy experienced by gay-male mixed status couples in home and public spheres. The impacts of HIV in the home were found to be minimalized on account of the HIV-positive partner's adherence to medications and achievement of an undetectable virus load. In public life, however, couples remained cautious about disclosing the status of their relationship to avoid stigma and discrimination [16]. While our study supports this research, we found that private normalization and public stigmatization of HIV were nevertheless connected to the construction of resilience in couples' everyday lives. Moreover, as Scambler has distinguished, there may be two different types of stigma, enacted (overt discrimination grounded in social unacceptability) and felt (shame and fear of confronting discrimination) stigma [24] co-exists in the everyday life of both partners of mixed-status relationships. Although enacted stigma has been gradually diminishing in the era of U = U and 'Treatment as Prevention', felt stigma has perpetuated in the social encounters of the mixed-status couples. For many, felt stigma was involved in their decision-making around disclosure to familial and social networks.

An absence of felt stigma was key to disclosure of HIV and further became a practice leading to actively seeking support from social networks outside of the relationship. Those networks then served to provide couples with improved access to HIV care and services, which in turn reinforced the capacity to achieve U = U and live "a normal life again". Indeed, participants' strategies to maintain an undetectable viral load and engender normalcy were anchored in rich, trusted, and sustainable resources within their social networks. We also saw that resilience was possible and achieved through diverse pathways that have not been previously

documented. In couples where the positive partner had already developed resilience to HIV, including strategies for reducing transmission risk, there appeared to be less energy devoted to managing HIV-related concerns and an easier pathway to normalcy. In couples where the positive partner was diagnosed after the start of a relationship, resilience was developed together, throughout a tense period where each partner came to terms with their individual concerns about transmission risk, management of HIV-related health challenges, and compassion for the HIV-positive partner's possibility of health impairment. For couples who remained together, this high-risk period facilitated a shared understanding of how to achieve resilience within the relationship.

Outside of relationships, we found disclosure was carefully managed. Governance over those decisions was universally accepted as laying with the HIV-positive partner, even if the other partner did contribute to the conversation by assessing levels of trust within their networks and the consequences of miscalculation, particularly stigma or conversely acceptance. The presence of advocates and gatekeepers in the HIV community within one's social network was crucial to these decisions. Participants connected to the GBMSM community enjoyed especially wider access to these supportive networks and interactions, enabling them to more easily disclose. The unique social capital of the GBMSM community, possibly stemming from their own history with resilience in the face of systemic stigmatization and adversity [25], has been suggested in other research with mixed-status couples [16].

Couples who avoided disclosure may have relinquished opportunities to access these supports which could ultimately have helped them to better navigate HIV-related challenges and build resilience to HIV in the context of their relationships. Indeed, interactions with health professionals and people working within AIDS-related community organizations, who belonged in some of our gay and/ bisexual participants' networks, helped to normalize HIV and its presence in intimate, sexual relationships. The networks also facilitated access to cutting-edge medical information and products supporting viral suppression and troubleshoot HIV-related concerns. The apparent plethora of ongoing HIV-related projects that GBMSM communities were exposed to also alleviated the pressure of disclosures; some participants secured needed HIV-related information and supports without having to formally disclose.

Disclosure as a driver of relationship resilience was, however, profoundly shaped by couples' social-cultural contexts. Participants who faced social precarity due to their legal or economic status, or lack of connection to a community that accepted or openly discussed HIV, actively avoided disclosure outside of their relationship for fear of stigmatization. Racialized persons and especially immigrants were more frequently represented in these narratives in our study. These findings reverberate with the findings of other studies conducted in Canada with racialized newcomers affected by HIV [26, 27]. Logie and colleagues suggest that couples in Canada who report difficulties in accessing healthcare services due to immigration, drug use, employment precariousness, incarceration, and other marginalizing circumstances require more intensive support from their family, friends, and local communities [9]. We saw how stigma and non-disclosure deterred access to such supports in exactly the couples who may need them the most.

In assessing the social determinants of HIV as they pertain to resilience among mixed-status couples, we uncovered critical gaps in access to HIV services in Canada. Despite enjoying a single-payer healthcare system, participants' narratives highlighted the challenges associated with the absence of a national drug plan. We saw how disparate provincial subsidies for allied health care, and disparate criteria for medication coverage [28] could disrupt access to HIV care continuity and equity. The challenge of scarce speciality resources in rural settings, changes to employment and health insurance, and relocation—that were reported by our participants—are consistent with prior work showing how economically disadvantaged persons

may be compelled to juggle adherence to HIV treatment and care with access to private insurance and ability to pay out-of-pocket expenditures [26, 29].

Our study had several limitations. Occasional discrepancies were noted in accounts of relationship wellbeing between partners in a relationship. These differences proved challenging to reconcile and might have reflected partners' attempt to protect the integrity of their relationship within an interview (social desirability bias). Alternatively, these discrepancies may have reflected authentic differences in how relationships were viewed by partners. More data are needed to bolster analysis of the lived experiences of socially marginalized groups in the Canadian context, such as Indigenous peoples and newcomers, to enhance the transferability of our analysis. Strengths of this study included the diversity of participants in our sample, by gender, sexual orientation, and geographic location, detailed accounts of lived relationship experiences which came through in our analysis, and the ability to interpret those experiences from a couples' resilience framework.

## Conclusion

In summary, we found that resilience was perceived among mixed-status couples as the capacity of living life as a normal couple, which was supported by HIV prevention paradigms linked to antiretroviral therapy such as TasP and U = U. We found that the pathways of constructing, shaping, and maintaining resilience among mixed-status couples were influenced by the temporality of HIV diagnosis i.e., the timing of HIV diagnosis in the relationship, access to high quality information and services, and the effects of disclosure, stigma, and social support. Couples who identified as GBMSM were able to rely on the social capitol of this community to maintain mutual wellbeing and resist disruptions to their relationships. Couples with fewer resources and more marginalized social identities faced greater barriers securing the resources required for building resilience. Future work dedicated to enhancing wellbeing among mixed-status couples should take stock of these experiences to better meet couples' needs in the design of accessible interventions, including in community setting, that could connect couples to peers and supportive services.

## Supporting information

**S1 Appendix. Appendix: Positive plus one study interview guides.** Qualitative interview guides.
(PDF)

**S1 File. Data availability and publication consent.**
(PDF)

## Acknowledgments

The authors wish to thank: all of the participants of the Positive Plus One study (613 from the initial survey, and the 51 purposively selected for the qualitative follow-up interviews utilised for this paper); the staff at the 178 service organizations and clinics from across Canada that supported the study through development and participant recruitment; and the study staff, collaborators, and investigators who assisted with study development and implementation.

## Author Contributions

**Conceptualization:** Minhui Yang, Amrita Daftary, Liviana Calzavara.

**Data curation:** Sandra Bullock.

**Formal analysis:** Minhui Yang, Molly Ryan.

**Funding acquisition:** Amrita Daftary, Joshua B. Mendelsohn, Laura Bisaillon, Adam Bourne, Tamara Thompson, Liviana Calzavara.

**Investigation:** Sandra Bullock, Liviana Calzavara.

**Methodology:** Amrita Daftary, Joshua B. Mendelsohn, Sandra Bullock, Liviana Calzavara.

**Project administration:** Sandra Bullock.

**Supervision:** Amrita Daftary, Joshua B. Mendelsohn, Sandra Bullock, Liviana Calzavara.

**Writing – original draft:** Minhui Yang, Amrita Daftary.

**Writing – review & editing:** Minhui Yang, Amrita Daftary, Joshua B. Mendelsohn, Molly Ryan, Sandra Bullock, Laura Bisaillon, Adam Bourne, Bertrand Lebouché, Tamara Thompson, Liviana Calzavara.

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
