## [Decision Letter · Decision Letter 0]

5 Aug 2022

PONE-D-22-15412‘"Live a normal life": Constructions of resilience among people in mixed HIV status relationships in CanadaPLOS ONE

Dear Dr. Sandra Bullock,

Thank you for submitting your manuscript to PLOS ONE. After careful consideration, we feel that it has merit but does not fully meet PLOS ONE’s publication criteria as it currently stands. Therefore, we invite you to submit a revised version of the manuscript that addresses the points raised during the review process.

We look forward to receiving your revised manuscript.

Kind regards,

Jeremiah Chikovore

Academic Editor

PLOS ONE

Journal Requirements:

“We thank the following: participants of the Positive Plus One study; and staff at the service organizations and clinics from across Canada that have supported the study through development and participant recruitment. The study was funded by the Canadian Institutes of Health Research (Grant # MOP-137009).”

“This research (all authors) was supported by the Canadian Institute of Health Research (CIHR, grant number: MOP-137009) and the Social Research Centre in HIV Prevention (SRC).

Funders' websites:

CIHR: https://cihr-irsc.gc.ca/e/193.html

SRC: https://srchiv.ca/en/index.php

4. PLOS requires an ORCID iD for the corresponding author in Editorial Manager on papers submitted after December 6th, 2016. Please ensure that you have an ORCID iD and that it is validated in Editorial Manager. To do this, go to ‘Update my Information’ (in the upper left-hand corner of the main menu), and click on the Fetch/Validate link next to the ORCID field. This will take you to the ORCID site and allow you to create a new iD or authenticate a pre-existing iD in Editorial Manager. Please see the following video for instructions on linking an ORCID iD to your Editorial Manager account: https://www.youtube.com/watch?v=_xcclfuvtxQ.

5. One of the noted authors is a group or consortium “the Positive Plus One Team”. In addition to naming the author group, please list the individual authors and affiliations within this group in the acknowledgments section of your manuscript. Please also indicate clearly a lead author for this group along with a contact email address.

Reviewers' comments:

Reviewer's Responses to Questions

**Comments to the Author**

1. Is the manuscript technically sound, and do the data support the conclusions?

Reviewer #1: Yes

Reviewer #2: Yes

2. Has the statistical analysis been performed appropriately and rigorously? 

Reviewer #1: N/A

Reviewer #2: N/A

3. Have the authors made all data underlying the findings in their manuscript fully available?

Reviewer #1: Yes

Reviewer #2: No

4. Is the manuscript presented in an intelligible fashion and written in standard English?

Reviewer #1: Yes

Reviewer #2: Yes

5. Review Comments to the Author

Reviewer #1: Thank you for your submission — “’Live a normal life’: Constructions of resilience among people in mixed HIV status relationships in Canada” — to PLOS ONE. I really enjoyed reading your manuscript and feel that it will make an excellent contribution to the field exploring how HIV impacts relationships among serodiscordant couples. I believe, however, that you can take a few opportunities to better structure your manuscript to better guide the reader to your conclusions (which I agree are supported by the data). I have structured my recommendations for each section below with suggestions for how to do so:

- Abstract

o It would be helpful to mention that you used thematic analysis to analyze the interviews in the abstract.

o It would also be helpful if you could define “normal couple” as conceptualized by participants, as you do later in the manuscript, in the abstract

o I recognize that these changes may be difficult within the word limits

- Introduction

o “Building on this work” should start a new paragraph to make it more clear that you are presenting the study question after it (it will just make it easier to skim and digest). You should also add another sentence briefly introducing the methods here “We used in-depth interviews to assess factors influencing resilience among XX mixed-status couples enrolled in Positive Plus One”

- Methods

o Consider adding section titles (recruitment, interviews, analysis plan) to make it easier to follow for the reader

o I had a difficult time figuring out if all the results from the qualitative interviews were presented in this analysis or only aspects relevant to resilience within couples.

o It may be helpful to add a table/supplement with interview questions to give readers a sense of what folks were asked.

o Good explanation of the inductive approach, this part of the methods may go better in its own section so it’s clearer when reading through

o It may be worth how many codes were generated within each theme, either in the methods or the results, so the reader can get a sense of the breath of ideas present in the interviews (if in the results can just go in the first sentence of each subsection)

- Results

o See the final comment from the methods, otherwise great work!

o I keep going back and forth on whether Pre- and Post-Disclosure need to be different themes, because it seems there is more anticipated stigma before disclosure and more experienced stigma (in case of that one PCP) and, thankfully more often, acceptance post-disclosure. Either way, it may at least be worth breaking down stigma a little bit into its different subtypes.

- Discussion

o I really like the conclusion paragraph, it’s a great summary of the paper! On the first reading, it was the first place that all that information came together for me, which I would have liked to have occurred while reading the results (or looking at the figure – which is poorly placed at the end for submissions, so maybe it would be different if I had seen it before reading the results section). I don’t really know what to do about this, but hopefully implementing some of the above suggestions will improve the structure and make this paper come together in a readers head before the end (although I don’t think this paragraph needs to change).

- Figure

o It may be worth considering placing which codes you generated within each theme on the figure

- General

o It would be helpful to see your codebook as a supplemental file or as a table

Thank you again for your submission!

Reviewer #2: Manuscript Number: PONE-D-22-04872

Manuscript Title: The dermatology clinic may be an effective human immunodeficiency virus (HIV) screening service delivery point for provider-initiated testing and counseling (PITC) services in Malawi

Summary

The authors describe resilience among HIV mixed status couples in Canada. Findings presented in this manuscript comprise a sub-analysis of qualitative data drawn from Positive Plus One Mixed methods study. Qualitative results contained in this manuscript are from an original research and have not been published elsewhere. There are however, a few areas that the authors should work on to further improve the manuscript. These areas have been described below:

Minor revisions

Abstract

Page 3: The abstract has information on methodology and findings. Authors should include a statement on key conclusion from this research. They should also provide information on how they analysed qualitative data.

Introduction

Page 4: Second paragraph: The authors describes that ‘For mixed HIV-status couples, resilience is found to be closely bound to HIV treatment and prevention.” This statement ignore the fact that ‘resilience’ within the context of mixed status couples is a complex process comprising other important elements that shapes its. I suggest that the authors should provide a description of other factors that influence resilience in mixed status couples in addition to HIV treatment and prevention. Examples of these may include level of trust, the differences in economic position and power between couples, cultural context among others.

Page 5: Second paragraph: A statement that begins with ‘Recent work from Australia suggests that couples… ‘ should read ‘Recent work from Australia suggests that HIV mixed status couples …. “

Methods

The methods section does not provide information on how study participants were approached i.e. face-to-face, telephone, email. There is also no statement that describes the number of individuals that refused to participate in the study and the reasons for this.

The authors have also not provided a theoretical framework that informed their methodology.

The authors have not provided the credentials of individuals that collected qualitative data - i.e. gender, sexual orientation, qualifications - and whether or not their attributes mirrors those of the study participants.

The authors should provide information on the setting where qualitative data was collected - home or clinic or a neutral venue.

Furthermore, there is need for authors to describe whether sexual partners where present during interviews.

Authors do not describe how they ensured that they collected sufficient data for this analysis.

Page 6: Second paragraph: Authors write - ‘Transcripts were coded by two research team members (MY, MR) who led the thematic analysis, used an inductive approach, identifying patterns within and across data to generate emerging themes.’ It is useful to indicate whether or not MY and MR were coding the data together or they performed parallel data coding followed by a coding consolidation meeting. The authors should also describe if data coding was done manually or with the help of a computer soft ware.

Results

It will be quite interesting of the analysis presented in the results section compares the experiences of resilience between sexual partiners that are living in mixed status heterosexual relationships to those living in mixed status GBSMS relationships. If the experiences were the same, it will be good that this should be mentioned.

Pages 7&8; Paragraph 1 and 2: Most of the qualitative studies include the demographic characteristics in the methods section and not in results section since participants selection is usually done using non-probabilistic sampling techniques. Author should provide an explanation on why they have included these demographic characteristics within the results section.

Page 8; Paragraph 3: Authors states ‘timing of HIV in the relationship.’ This phrase has been presented in several sections of the manuscript. This phrase need to be clarified. Do they refer to the ‘timing of entry of HIV within the relationship’ or ‘the timing of learning that they are living in a mixed status relationship.’

Page 8; The theme titles ‘ feeling and being ‘normal’; entering intimate relationships, accessing information and services etc can be creatively presented. For example, access to reliable medical information and healthcare services; status disclosure, HIV stigma and acceptance etc. I think that themes presented in Figure 1 can been better sub-titles for themes of this manuscript.

Sub-title : Assessing information

Page 13. Paragraph 1: Replace the word ‘discordance’ with ‘mixed status’ or sero different’

Page 14. Authors states ‘Financial barriers, for example, were only partly alleviated by the public health system that offered basic HIV treatment services at no charge but limited access to allied healthcare services.” It is useful to mention a few of these ‘allied health care services’.

Quotes in the results section: It is useful to provide a much richer description of the characteristics of the individuals who provided these quotes. Eg. HIV positive heterosexual man, 30 years, employed, XX province.

Discussion

Page 23: Authors write “We uncovered how pharmaceutical technologies,…” In the result section, there is no mention of pharmaceutical technologies. It will therefore be important to indicate the names of these technologies which have also been presented in the results section.

6. PLOS authors have the option to publish the peer review history of their article (what does this mean?). If published, this will include your full peer review and any attached files.

Reviewer #1: No

Reviewer #2: No

---

## [Author Response · Author response to Decision Letter 0]

23 Nov 2022

Response to Reviewers

Response to Editor’s comments: 

Thank you for your reminder on formatting. I believe we have met all required formatting requirements. 

“We thank the following: participants of the Positive Plus One study; and staff at the service organizations and clinics from across Canada that have supported the study through development and participant recruitment. The study was funded by the Canadian Institutes of Health Research (Grant # MOP-137009).”

“This research (all authors) was supported by the Canadian Institute of Health Research (CIHR, grant number: MOP-137009) and the Social Research Centre in HIV Prevention (SRC).

Funders' websites:

CIHR: https://cihr-irsc.gc.ca/e/193.html

SRC: https://srchiv.ca/en/index.php

Thank you for the clarification regarding the funding and acknowledgements statements. 

• We have removed all funding related information from the Acknowledgements section. It now reads: 

o Page 36, Line 742 - 746: The authors wish to thank: all of the participants of the Positive Plus One study (613 from the initial survey, and the 51 purposively selected for the qualitative follow-up interviews utilised for this paper); the staff at the 178 service organizations and clinics from across Canada that supported the study through participant recruitment; and the investigators and collaborators who assisted with study development.

• The updated funding statement has been provided in the Cover Letter.

o This research was supported by the Canadian Institute of Health Research (CIHR operating grant number: MOP-137009) and the Social Research Centre in HIV Prevention (SRC). The funders had no role in study design, data collection and analysis, decision to publish, or preparation of the manuscript.

Thank you for this clarification. The required ethics statement has been updated in the “Methods” section, “Research ethics” subsection of the manuscript. 

• Pages 10 - 11, Lines 189 - 211: All methods for this study were carried out in accordance with relevant guidelines and regulations. This study received ethics approval from the University of Toronto research ethics board (REB) (Protocol 31855). Given involvement in active recruitment, yet no involvement in participant consent or data collection procedures, only a few of the health/HIV clinics judged it necessary to obtain approval from their own REB. The study underwent review and obtained approval from REBs at McGill University (2017-1779, 16-035-MUHC, eReviews_5368), University of Saskatchewan (15-399), St. Michael's Hospital (16-343), Toronto Public Health (2016-02), Nova Scotia Health Authority (NSHA REB ROMEO FILE #: 1022121), Prince Albert Parkland Health Region (no REB number), and Regina Qu’Appelle Health Region (REB-15-133). 

All research team members and staff signed an oath of confidentiality. All participants indicated their informed consent to participate in the initial survey following completion of eligibility screening – either verbally if completed by telephone, or by checking the “I consent” box if completed online. The survey program did not retain screening data if consent was not provided. Further telephone, verbal consent was required for participants to complete the in-depth qualitative interview, providing permission to publish anonymous quotes from their interviews. Consent of both partners within an HIV mixed-status relationship was required to link their responses; and participants’ survey responses were not shared with their partners even when their data was linked for analysis. Survey and qualitative data were anonymous, and unlinked contact information was provided to receive a token of appreciation and to volunteer for the qualitative interview; and was deleted once it was no longer required for these purposes. All anonymized data will be kept for seven years post project completion.

4. PLOS requires an ORCID iD for the corresponding author in Editorial Manager on papers submitted after December 6th, 2016. Please ensure that you have an ORCID iD and that it is validated in Editorial Manager. 

• The corresponding author -- Sandra Bullock’s ORCID is https://orcid.org/0000-0002-4913-5745 ; this has been validated in the Editorial Manager.

5. One of the noted authors is a group or consortium “the Positive Plus One Team”. In addition to naming the author group, please list the individual authors and affiliations within this group in the acknowledgments section of your manuscript. Please also indicate clearly a lead author for this group along with a contact email address.

Thank you for clarifying the requirements for using consortiums as authors. The “Positive Plus One Team” author designation was meant to be inclusive of all of our investigators and collaborators as well as the over 100 recruitment sites. Neither collaborators nor recruiters were individuals, they were organizations; the people we worked with came and went over time due to staff turnover, and many organizations no longer exist due to Canadian federal funding cuts through 2017 and 2018. So names, affiliations, and contact info are not available for these. Therefore, we have now done the following:

• Page 1, Authors: We have eliminated the “Positive Plus One Team” from the author list and refer to these parties in the Acknowledgements statement only.

• Page 36, Lines 742 – 746, Acknowledgements: The following statement replaces the “positive Plus One Team” author designation.

The authors wish to thank: the participants of the Positive Plus One study (613 from the initial survey, and the 51 purposively selected for the qualitative follow-up interviews utilised for this paper); the staff at the 178 service organizations and clinics from across Canada that supported the study through development and participant recruitment; and the study staff, collaborators, and investigators who assisted with study development and implementation.

Thank you for your comment regarding data availability. These study data are not available via open access. In order to ensure openness in responses from both partners in a relationship dyad, potential participants were assured in the informed consent process that we would not divulge one partner’s responses to the other partner. All participants agreed to the following “Although we may use quotes from your interview in future publications or presentations, all quotes will be anonymous. Also, each quote will be general enough in nature that you will not be identifiable, even by your partner.” At the time we did not request permission from participants to make their data available via open access.

By making full, linked interviews available via open access we would risk violating that condition of consent. Further, because of the nature of this qualitative inquiry, even deidentified interviews will hold information (e.g., storylines) that could breach participant confidentiality and anonymity around a very sensitive subject area. Prior to allowing other researchers access to the data they would be required to uphold this condition. Therefore, as per the journal’s policy, there are clear ethical restrictions regarding publicly sharing these data, and request a waiver of this requirement. 

• In cover Letter, and in the file titled “data availability and publication consent” and uploaded as a “supporting information” document: Our data availability statement should read: 

De-identified data excerpts are presented within the paper, as a minimal dataset. There are ethical restrictions on sharing the complete de-identified study transcripts, due to the ability of one person to identify their partner’s statements. The study consent letter included the statement: “Although we may use quotes from your interview in future publications or presentations, all quotes will be anonymous. Also, each quote will be general enough in nature that you will not be identifiable, even by your partner.” Providing fully-transcribed interviews would allow one partner to easily identify their partner, and possibly other people. As a result, the data can only be made available to other researchers if terms of anonymity are met via data transfer agreements with the corresponding author. 

Responses to Reviewers’ comments:

Reviewer #1: Thank you for your submission — “’Live a normal life’: Constructions of resilience among people in mixed HIV status relationships in Canada” — to PLOS ONE. I really enjoyed reading your manuscript and feel that it will make an excellent contribution to the field exploring how HIV impacts relationships among serodiscordant couples. I believe, however, that you can take a few opportunities to better structure your manuscript to better guide the reader to your conclusions (which I agree are supported by the data). I have structured my recommendations for each section below with suggestions for how to do so:

Thank-you for your kind evaluation of our manuscript. We thank you for your time and effort to review it and for the valuable feedback provided. Page and line numbers listed below refer to the revised manuscript as seen in “Simple markup” track changes.

1. Abstract

It would be helpful to mention that you used thematic analysis to analyze the interviews in the abstract. It would also be helpful if you could define “normal couple” as conceptualized by participants, as you do later in the manuscript, in the abstract. I recognize that these changes may be difficult within the word limits

Thank you for these comments. We have added further information regarding the analytical methods to sentence 2 of the abstract, and detailed the meaning of a “normal couple” in sentence 3.

• Page 3, Lines 28 - 30: Qualitative interviews with 51 participants (10 women, 41 men, including 27 HIV-positive and 24 HIV-negative partners) were analyzed using inductive thematic analysis to examine notions of relationship resilience in the context of emerging HIV social campaigns. 

• Page 3, Lines 31 - 33: Relationship resilience meant finding ways to build and enact life as a normal couple, that is, the couples that have not been noticeably affected by HIV, linked to the partner with HIV maintaining viral suppression and achieving U=U (undetectable=untransmittable).

2. Introduction

“Building on this work” should start a new paragraph to make it more clear that you are presenting the study question after it (it will just make it easier to skim and digest). You should also add another sentence briefly introducing the methods here “We used in-depth interviews to assess factors influencing resilience among XX mixed-status couples enrolled in Positive Plus One”

We have adjusted the introduction to reflect these comments.

• Page 5, Line 76: now begins with “Building on this work…” 

• Page 5, Lines 78 - 80: we have added the sentence “To achieve this goal, we used in-depth interviews to assess factors influencing resilience among 51 people from 32 mixed-status couples enrolled in the Positive Plus One study.”

2. Methods

a) Consider adding section titles (recruitment, interviews, analysis plan) to make it easier to follow for the reader.

Thank you for the suggestion, we have added these level 2 headings. Please note that the use these headings has led to some rearrangement of the Methods section as a whole.

• Page 6, Line 93: Recruitment

• Page 7, Line 115: Study participants

• Page 8, Line 138: The interview

• Page 9, Line 155: Analysis plan

• Page 10, Line 189: Research ethics

b) I had a difficult time figuring out if all the results from the qualitative interviews were presented in this analysis or only aspects relevant to resilience within couples.

Thank you for bringing this to our attention. In the Methods section, we have clarified that this paper only focuses on a part of the results from a larger project that included both an initial survey component and a follow-up qualitative in-depth interview. The results presented here are only from the purposively selected sub-sample of those who volunteered to participate in the in-depth interview and include only qualitative results pertaining to resilience. Other results, quantitative (citations 14, 17, 18) and qualitative (citation 11) have been published elsewhere, and have been duly cited. This paper includes analysis of all 51 qualitative interviews. 

• Page 6, Lines 91 – 92: This is one of two qualitative papers from the study, and focuses solely on the results pertaining to resilience. 

• Page 3, Lines 28 – 39, Abstract: Qualitative interviews with 51 participants (10 women, 41 men, including 27 HIV-positive and 24 HIV-negative partners) …

• Page 10, Line 175: In the first round of reading and coding of the 51 interviews, we identified concepts related to …

c) It may be helpful to add a table/supplement with interview questions to give readers a sense of what folks were asked.

Thank you for this suggestion. We have provided the interview schedules as a “supporting information” document to accompany the paper. It is titled: “Qualitative Interview Guides”. 

d) Good explanation of the inductive approach, this part of the methods may go better in its own section so it’s clearer when reading through. It may be worth how many codes were generated within each theme, either in the methods or the results, so the reader can get a sense of the breath of ideas present in the interviews (if in the results can just go in the first sentence of each subsection)

Thank for the suggestion; we have created a new Methods sub-section “Analysis plan” to clarify the procedures for processing and coding the interviews. A total of 118 codes were identified and used in the preparation of this paper.

• Page 10, Lines 175 - 180: In the first round of reading and coding of the 51 interviews, we identified concepts related to sexual and intimate relationships (18 codes), HIV disclosure (9 codes), family relations (5 codes), experiences in health care (15 codes) , diagnosis (6 codes), medication (9 codes), transmission (7 codes), social support (21 codes), knowledge and attitudes about HIV (17 codes), and lived experience (11 codes) as meaningful aspects of mixed-status relationships for study participants. 

3. Results. See the final comment from the methods, otherwise great work! I keep going back and forth on whether Pre- and Post-Disclosure need to be different themes, because it seems there is more anticipated stigma before disclosure and more experienced stigma (in case of that one PCP) and, thankfully more often, acceptance post-disclosure. Either way, it may at least be worth breaking down stigma a little bit into its different subtypes.

Thank you for the vote of confidence and we appreciate the suggestion to develop separate themes around pre- and post-disclosure. Instead we have chosen to retain the umbrella theme about disclosure to ensure the nuance and non-linearity of couples’ relationships are maintained for the reader, and because we have further discussed ideas of felt and enacted stigma that we choose not to distinguish into separate themes but rather enrich the one theme of disclosure. This is because in separating, there is an implicit hierarchy sometimes given to the issue of felt stigma relative to enacted stigma (as if felt stigma is not as relevant) but indeed in our study both types of stigma were meaningful and impactful for couple’s resilience. We have added several sentences in the Results and Discussion sections to clarify the stigma experienced by the mixed-status couples. 

• Page 26, Lines 539 - 542: However, despite the ongoing public normalization of HIV enabled by the new paradigms of U=U and treatment as prevention, stigma and discriminations to PLHIV perpetuated in some micro interpersonal environments. Couples who felt this stigma in their close circles tended to avoid …

• Pages 28, Lines 566 – 572: Moreover, as Scambler has distinguished, there may be two different types of stigma, enacted (overt discrimination grounded in social unacceptability) and felt (shame and fear of confronting discrimination) ones co-exist in the everyday life of both partners of the mixed-status relationships. Although enacted stigma has been gradually diminishing in the era of U=U and ‘Treatment as Prevention’, the felt stigma was perpetuated in the social encounters of the mixed-status couples. For many, the felt stigma was involved in their decision-making around disclosure to familial and social networks.

4. Discussion. I really like the conclusion paragraph, it’s a great summary of the paper! On the first reading, it was the first place that all that information came together for me, which I would have liked to have occurred while reading the results (or looking at the figure – which is poorly placed at the end for submissions, so maybe it would be different if I had seen it before reading the results section). I don’t really know what to do about this, but hopefully implementing some of the above suggestions will improve the structure and make this paper come together in a readers head before the end (although I don’t think this paragraph needs to change).

Thank you very much for this comment. In order to help the reader follow the building of the model that is represented by Figure 1, we have moved the first paragraph of the Discussion, that is the summary of our model, to the Results section. Placing this information earlier helps to put the results sections more into context and describe the inter-relationship between the constructs. 

• Page 11 - 12, Lines 214 - 223: This research occurred during a period of rapid acceptance of messaging centered on U=U in Canada, a concept that has reshaped perceptions of HIV and associated stigma but, as our study shows, is not yet universal or sufficient to quell concerns among all groups of people. In-depth interviews with participants engaged in mixed-status relationships across the country point to a shared understanding of resilience as the capacity of keeping life as normal. We uncovered how pharmaceutical technologies (i.e., current ART for achieving “treatment as prevention,” and the use of pre-exposure prophylaxis), access to information and services, ease of disclosure, and social acceptance, both enabled by and enabling achievement of U=U. These factors contribute to shaping the constructions of normalcy and, in turn, not only affirm couples’ resilience but also serve as a gateway for expanded relationship resilience. 

5. Figure. It may be worth considering placing which codes you generated within each theme on the figure

Thank you. To keep the figure clear and closely connected to main findings (main storyline), we do not wish to add all codes into the figure. However, we have included a list of codes and frequencies in the last paragraph of method section to reflect the richness and depth of our analysis. Please see our response to reviewer 1, comment 2d. 

6.General

o It would be helpful to see your codebook as a supplemental file or as a table

Please see our response to comment 2d. The full codebook is connected to other analyses as well, some of which have been published (e.g., Molly Ryan et al., 2021), and hence not shared. 

Reviewer #2: Manuscript Number: PONE-D-22-15412

Manuscript Title: “’Live a normal life’: Constructions of resilience among people in mixed HIV status relationships in Canada” 

Summary

The authors describe resilience among HIV mixed status couples in Canada. Findings presented in this manuscript comprise a sub-analysis of qualitative data drawn from Positive Plus One Mixed methods study. Qualitative results contained in this manuscript are from an original research and have not been published elsewhere. There are however, a few areas that the authors should work on to further improve the manuscript. These areas have been described below:

Minor revisions

Abstract

Page 4, : The abstract has information on methodology and findings. Authors should include a statement on key conclusion from this research. They should also provide information on how they analysed qualitative data.

Thank you for the comment. We have edited the abstract to address these issues. 

We have added a line to indicate how the data were analysed.

• Page 3, Lines 28 - 30: Qualitative interviews with 51 participants (10 women, 41 men, including 27 HIV-positive and 24 HIV-negative partners) were analyzed with inductive thematic analysis to examine notions of relationship resilience in the context of emerging HIV social campaigns.

The addition of this sentence provides a concluding statement to the abstract

• Page 3, Lines 38 – 40: We conclude that important pathways of constructing, shaping, and maintaining resilience were influenced by the timing of HIV entry into the relationship, access to HIV-related information and services, disclosure, stigma, and social acceptance.

Introduction

Page 4: Second paragraph: The authors describes that ‘For mixed HIV-status couples, resilience is found to be closely bound to HIV treatment and prevention.” This statement ignore the fact that ‘resilience’ within the context of mixed status couples is a complex process comprising other important elements that shapes its. I suggest that the authors should provide a description of other factors that influence resilience in mixed status couples in addition to HIV treatment and prevention. Examples of these may include level of trust, the differences in economic position and power between couples, cultural context among others.

Thank you for highlighting this important point. We had indicated in the second paragraph of the introduction section that “resilience can have fluid and multi-dimensional constructions, depending on the cultural and material contexts in which it is developed” prior to the mentioned sentence. However, we also have identified some of the complexities of resilience to this paragraph to address your concerns.

• Page 4, Lines 59-62: Resilience can have fluid and multi-dimensional constructions, depending on the cultural and material contexts in which it is developed; such as: level of trust, the differences in economic position and power between couples, cultural understandings about HIV and about relationships.5,9,10 

Page 5: Second paragraph: A statement that begins with ‘Recent work from Australia suggests that couples… ‘ should read ‘Recent work from Australia suggests that HIV mixed status couples …. “

Thank you very much for the suggestion. We have edited this sentence accordingly. 

• Page 5, Lines 72 - 74: Recent work from Australia suggests that HIV mixed-status couples may also face distinct experiences within their home as compared to the public domain.16

Methods

The methods section does not provide information on how study participants were approached i.e. face-to-face, telephone, email. 

Thank you for this comment. We have now provided further information to address the approach and recruitment of participants. 

It should be noted that this qualitative analysis involved a sub-sample of a larger study. We have added limited information on recruitment of the 613 participants of the larger Canadian national sample, and refer readers to our methodological paper published in BMC – Public Health (citation 18) for complete details on the larger study. It is now found at the beginning of the “Recruitment” sub-heading.

• Page 6, Lines 94 – 97: Participants were recruited via 143 AIDS Service Organizations and other non-governmental organizations and 35 Infectious disease and general health clinics, via direct contact which staff and physicians, and indirectly via poster and pamphlets, word-of-mouth, media articles, and social media.

Further detail has been provided on how participants were then introduced, and invited to volunteer for the qualitative in-depth study, within the same paragraph. 

• Page 6, Lines 102 – 106: After completion of the survey, 51.7% of the participants indicated interest in volunteering for an in-depth audio-recorded telephone interview. Interested participants were asked to provide their contact information – email was preferred by 54%; phone by 29%; and 17% indicated both or either. For the 17% who indicated either contact method, three attempts were made by telephone prior to contacting their email addresses. 

There is also no statement that describes the number of individuals that refused to participate in the study and the reasons for this. 

Thank you for bringing this to our attention. We do not have reasons for refusal from the main survey as it was a convenience sample and people only initiated the online survey or called the office for a telephone survey if they were willing to participate. Since over 300 individuals volunteered for the in-depth interview and we were interested in obtaining a diverse subsample of those volunteering, we only invited a subset of the volunteers to take part. We have added additional information to the Methods section, Recruitment subsection, to reflect our process and refusals. 

• Page 6, Lines 98 – 102: Survey participation was voluntary, people entered the online web survey, or called the office to take-part by telephone, only if they were interested in taking part in the study; therefore, we have no knowledge of the number people in mixed HIV-status relationships choosing not to participate, nor their reasons for non-participation. 

• Page 7, Lines 110 – 114: Of the volunteers, 69 were purposively selected for interview; 51 were ultimately interviewed prior to reaching data saturation. Two people (3%) were no longer interested – they reported being too busy; and 16 people (23%) were unable to be contacted/located by the time we were reaching out by email and telephone to schedule interviews, prior to reaching data saturation. 

The authors have also not provided a theoretical framework that informed their methodology.

Thank you. We approached this analysis with a resilience framework in mind as introduced in the introduction. We have now made the application of this as a theoretical framework to inform analysis more clear in the “Analysis plan” sub-section of the Methods. 

• Page 9, Lines 158 - 159: Due to the exploratory nature of the study, we chose an inductive thematic analysis methodology for the data analysis.

• Page 9, Lines 160 - 166: We concurrently picked up on coping strategies that mixed-status couples engage in to take care of themselves. This led us to adopt resilience as a theoretical basis from which to enhance our interpretations of data pertaining to couple’s means of dealing with HIV and maintaining mutual wellbeing in the contexts of their relationships. Accordingly, we refined our interpretations around inductively emergent codes and concepts with published frameworks of resilience applied to understand experiences of PLHIV as well as couples affected by HIV. 

The authors have not provided the credentials of individuals that collected qualitative data - i.e. gender, sexual orientation, qualifications - and whether or not their attributes mirrors those of the study participants.

Thank you for addressing this point. We have added a sentence to “The interview” sub-section of the Methods to indicate the backgrounds of the interviewers. 

• Page 8, Lines 144 - 148: The four interviewers were selected to represent key demographic groups of participants. One interviewer was HIV-positive and had previously been in an HIV mixed-status relationship; three were female, one male; two identified as heterosexual, one bisexual, and one was gay; and each of them had worked in the area of HIV research for several years.

The authors should provide information on the setting where qualitative data was collected - home or clinic or a neutral venue. Furthermore, there is need for authors to describe whether sexual partners where present during interviews.

Thank you for this query. We have added related information in the Methods section, “The interview” sub-section. 

• Page 8, Lines 139 - 144: Data were obtained by telephone interview. When speaking with the participant to set the appointment for the interview, participants were advised to ensure that they would be in a private location where no one was able to overhear and when their partner would not be present. Prior to obtaining verbal informed consent, interviewers asked participants if they were alone in a private location where they would not be overheard or interrupted. Informed consent was obtained prior to interview start. 

Authors do not describe how they ensured that they collected sufficient data for this analysis.

We agree with the reviewer and have added sentences in the Methods section (under sub-title “The interview”) to describe how we reached data saturation. 

• Page 8 - 9, Lines 150 - 154: To assess data saturation, the research coordinator listened to each interview as it was returned to the office following completion to assess for repeated responses to questions. Interviewers also met with the research coordinator biweekly to review enrollment, discuss emergent themes, and discuss data saturation. Data collection was completed once all parties agreed that responses were routinely repetitive, and no new information was presented.

Page 6: Second paragraph: Authors write - ‘Transcripts were coded by two research team members (MY, MR) who led the thematic analysis, used an inductive approach, identifying patterns within and across data to generate emerging themes.’ It is useful to indicate whether or not MY and MR were coding the data together or they performed parallel data coding followed by a coding consolidation meeting. The authors should also describe if data coding was done manually or with the help of a computer soft ware.

Thank you so much for the suggestions. We have edited this paragraph accordingly. 

• Page 9, Lines 170 – 171: MY and MR performed parallel data coding using Dedoose desktop software followed by meetings to consolidate coding.

Results

It will be quite interesting of the analysis presented in the results section compares the experiences of resilience between sexual partners that are living in mixed status heterosexual relationships to those living in mixed status GBMSM relationships. If the experiences were the same, it will be good that this should be mentioned.

Thank you for this comment. In the Results section, we highlighted the distinct experience of couples who identified as gay etc., (see page 12, Lines 234 – 236; and page 24, Lines 492 – 500). We also emphasize these critical experiences in paragraph 4 and 5 (pages 29 – 30) of the Discussion. However, the richness of narratives about the social capital and network available to gay couples was much higher in our study, than the lack of the same richness among heterosexual couples; a fair direct comparison was thus less feasible. However, based on the reviewer’s suggestion, we have highlighted this point now in the conclusion (page 32, lines 656 – 657). 

• Page 12, Lines 234 – 236: Noticeably, compared with heterosexual couples, we found that gay couples were more likely to obtain support from local LGBTQ communities and corresponding resources.

• Page 32, Lines 656 – 657: Couples who identified as GBMSM were able to rely on the social capital of this community to maintain mutual wellbeing and resist disruptions to their relationships.

Pages 7&8; Paragraph 1 and 2: Most of the qualitative studies include the demographic characteristics in the methods section and not in results section since participants selection is usually done using non-probabilistic sampling techniques. Author should provide an explanation on why they have included these demographic characteristics within the results section.

Thank you for this comment. We have moved the demographic characteristics of the sample to a new Methods sub-section titled “Study participants.” Here we have put together all descriptive information, including demographics on the 51 qualitative interview participants. 

• Pages 7 - 8, Lines 115 – 137: Newly compiled “Study participants” section. 

Page 8; Paragraph 3: Authors states ‘timing of HIV in the relationship.’ This phrase has been presented in several sections of the manuscript. This phrase need to be clarified. Do they refer to the ‘timing of entry of HIV within the relationship’ or ‘the timing of learning that they are living in a mixed status relationship.’

Thank you for this comment. We have changed the phrase “the timing of HIV in the relationship” to “the timing of HIV diagnosis within the relationship”. This has been changed in several locations: 

• Pages 3, 12, 15, 23

Page 8; The theme titles ‘ feeling and being ‘normal’; entering intimate relationships, accessing information and services etc can be creatively presented. For example, access to reliable medical information and healthcare services; status disclosure, HIV stigma and acceptance etc. I think that themes presented in Figure 1 can been better sub-titles for themes of this manuscript.

Sub-title : Assessing information

Thank you for this suggestion. We attempted to maintain the future as simple and clear as possible, however, the nuances around some ideas were better explicated in-text using slightly different language in some instances. We have elsewhere now aligned the sub-headings to be more descriptive and consistent with the figure terms. The revised Results section, level-2 headings follow

Page 13:

From: Feeling and being “normal” 

To: Resilience: Capacity to achieve U=U and feel “normal”

Page 15:

From: Entering intimate relationships

To: Timing of HIV diagnosis and entering intimate relationships 

Page 17:

From: Accessing reliable information and services

To: Accessing reliable HIV information and services

Page 22:

From: Disclosing, being stigmatized and accepting

To: Disclosing HIV status, being stigmatized or accepted

Page 13. Paragraph 1: Replace the word ‘discordance’ with ‘mixed status’ or sero different’

Thank you. We have replaced the “discordance” with “mixed-status”. The term “serodiscordant” remains only in the References section. 

Page 14. Authors states ‘Financial barriers, for example, were only partly alleviated by the public health system that offered basic HIV treatment services at no charge but limited access to allied healthcare services.” It is useful to mention a few of these ‘allied health care services’.

Thank you for the suggestion. We have added some examples of allied healthcare services in the text.

• Page 18, Lines 361 – 364: “Financial barriers, for example, were only partly alleviated by the public health system that offered basic HIV treatment services at no charge but limited access to allied healthcare services such as prescription medication, physiotherapy, mental health and addiction services, etc.”

Quotes in the results section: It is useful to provide a much richer description of the characteristics of the individuals who provided these quotes. Eg. HIV positive heterosexual man, 30 years, employed, XX province.

Thank you for this comment. We initially considered providing more descriptive details for each participant. However, given the obligation to protect participant identities including from their sexual partners who might have known of study participation and could potentially be able to identify their partners if multiple identifiers are given, we elected to limit these identifiers. 

Discussion

Page 23: Authors write “We uncovered how pharmaceutical technologies,…” In the result section, there is no mention of pharmaceutical technologies. It will therefore be important to indicate the names of these technologies which have also been presented in the results section.

Thank you for the suggestion. We have moved this paragraph up to the first paragraph of the Results section (page 11 -12, line 214 - 223), and added a note in parentheses to explain what the “pharmaceutical technologies” refers to. 

• Page 12, Line 219 - 220: “We uncovered how pharmaceutical technologies (i.e., current ART for achieving “treatment as prevention”, and the use of pre-exposure prophylaxis), …”

---

## [Decision Letter · Decision Letter 1]

20 Jan 2023

‘"Live a normal life": Constructions of resilience among people in mixed HIV status relationships in Canada

PONE-D-22-15412R1

Dear Dr. Bullock,

We’re pleased to inform you that your manuscript has been judged scientifically suitable for publication and will be formally accepted for publication once it meets all outstanding technical requirements.

Kind regards,

Jeremiah Chikovore

Academic Editor

PLOS ONE

Additional Editor Comments (optional):

Congratulation on the well-worked revisions to your paper. The reviewers have reverted with minor discretional comments that I would like to request you to consider.

Reviewers' comments:

Reviewer's Responses to Questions

**Comments to the Author**

1. If the authors have adequately addressed your comments raised in a previous round of review and you feel that this manuscript is now acceptable for publication, you may indicate that here to bypass the “Comments to the Author” section, enter your conflict of interest statement in the “Confidential to Editor” section, and submit your "Accept" recommendation.

Reviewer #1: All comments have been addressed

Reviewer #2: All comments have been addressed

2. Is the manuscript technically sound, and do the data support the conclusions?

Reviewer #1: Yes

Reviewer #2: Yes

3. Has the statistical analysis been performed appropriately and rigorously? 

Reviewer #1: N/A

Reviewer #2: N/A

4. Have the authors made all data underlying the findings in their manuscript fully available?

Reviewer #1: Yes

Reviewer #2: No

5. Is the manuscript presented in an intelligible fashion and written in standard English?

Reviewer #1: Yes

Reviewer #2: Yes

6. Review Comments to the Author

Reviewer #1: Thank you for your resubmission — “’Live a normal life’: Constructions of resilience among people in mixed HIV status relationships in Canada” — to PLOS ONE. I appreciate your thoughtful responses to my and the other reviewer’s comments and feel that you have sufficiently improved the manuscript to warrant publication. Thank you for your contribution to the field! Prior to publication, however, I feel there are some minor (voluntary) tweaks that can further improve the manuscript and, whether or not you implement them, I do not need to see this again to feel comfortable it being published:

- Abstract

o Line 32: instead of “the couples that have not been noticeably affected by HIV” you could write, “a couple not noticeably affected by HIV”

- Introduction

o Line 59: do you mean to start with “Resilience within couples” instead of “Resilience” or even, “Resilience, particularly within couples, can…” Since I believe you are focusing on resilience within couples, which is similar but not exactly the same as resilience in general (and you’re focusing in on resilience within couples).

- Methods

o Recruitment Section: Since you have survey data from interviewed participants, it may be worth comparing demographic data for participants who you couldn’t reach to participant versus those who participated (if you can do so safely without identifying folks).

o Study Participants: I know the other reviewer asked you to describe the participants in the methods…but I personally prefer this to go into the results. I believe this likely a training-specific preference and I don’t think you need to change it, but if you prefer that way too you have my support! Of note, this could be the section where you describe those who you couldn’t reach to participate. Either way, it would help to also have this information presented as a table.

o The Interview: Maybe remove “The” from the section header.

o Analysis Plan: how many times were interviews translated/back translated?

- Results

o None

- Discussion

o None

- Figure

o None

- General

o Great work, it was a pleasure to read!

Reviewer #2: I am satisfied that authors of this manuscript have addressed all the comments that I raised. I think that the paper in its current form is ready to be published. However, the issue of data availability needs special consideration as the authors indicated that data such as participant transcripts cannot be shared publicly. This is because they want to maintain the confidentiality of dyad partners and this is in line with the conditions of the consent.

7. PLOS authors have the option to publish the peer review history of their article (what does this mean?). If published, this will include your full peer review and any attached files.

Reviewer #1: No

Reviewer #2: **Yes: **Moses Kelly Kumwenda

---

## [Editor Report · Acceptance letter]

27 Feb 2023

PONE-D-22-15412R1 

“Live a normal life”: Constructions of resilience among people in mixed HIV status relationships in Canada 

Dear Dr. Bullock:

I'm pleased to inform you that your manuscript has been deemed suitable for publication in PLOS ONE. Congratulations! Your manuscript is now with our production department. 

Kind regards, 

on behalf of

Dr. Jeremiah Chikovore 

Academic Editor

PLOS ONE